# Bio-Computational Evaluation of Compounds of Bacopa Monnieri as a Potential Treatment for Schizophrenia

**DOI:** 10.3390/molecules27207050

**Published:** 2022-10-19

**Authors:** Ali H. Alharbi

**Affiliations:** Department of Health Informatics, College of Public Health and Health Informatics, Qassim University, Al Bukayriyah 52741, Saudi Arabia; ahhrbie@qu.edu.sa

**Keywords:** schizophrenia, neuronal disorders, STXBP1, *Bacopa monnieri*, MDS

## Abstract

Schizophrenia is a horrible mental disorder characterized by distorted perceptions of reality. Investigations have not identified a single etiology for schizophrenia, and there are multiple hypotheses based on various aspects of the disease. There is no specific treatment for schizophrenia. Hence, we have tried to investigate the updated information stored in the genetic databases related to genes that could be responsible for schizophrenia and other related neuronal disorders. After implementing combined computational methodology, such as protein-protein interaction analysis led by system biology approach, in silico docking analysis was performed to explore the 3D binding pattern of *Bacopa monnieri* natural compounds while interacting with STXBP1. The best-identified compound was CID:5319292 based on −10.3 kcal/mol binding energy. Further, selected complexes were dynamically evaluated by MDS methods, and the output reveals that the STXBP1-CID:5281800 complex showed the lowest RMSD value, i.e., between 0.3 and 0.4 nm. Hence, identified compounds could be used to develop and treat neuronal disorders after in vivo/in vitro testing.

## 1. Introduction

Schizophrenia is a horrible mental disorder characterized by distorted perceptions of reality. It can produce hallucinations, delusions, and profoundly disordered thought and behavior, impair daily functioning and be catastrophic. Patients with schizophrenia require lifelong care; early intervention may help manage symptoms before severe complications occur and improve the prognosis (https://www.webmd.com/schizophrenia/mental-health-schizophrenia (accessed on 22 August 2022)).

Approximately 24 million globally, or 1 in 300 persons, have schizophrenia (0.32 percent). Adults had a rate of 1 in 222 (or 0.45%) during this era. It does not occur as frequently as many other mental illnesses. The most common times for onset are in late adolescence and the early twenties, and onset often occurs earlier in men than in women. Investigations have not identified a single etiology for schizophrenia. It is theorized that genetic and environmental factors work together to generate schizophrenia. Psychosocial variables may also influence the onset and progression of schizophrenia. The use of cannabis is associated with an elevated risk for the disease.

There are multiple hypotheses based on various aspects of the disease, some of which are attributable to very well-known mechanisms of treatment therapies. The most widely accepted neurodevelopmental theory of schizophrenia combines environmental factors and causal genes. The dopamine hypothesis is based on the fact that all conventional therapies involve anti-dopaminergic processes, and genes such as DRD2, DRD3, DARPP-32, BDNF, or COMT are linked to dopaminergic system function [1]. The glutamatergic theory has recently resulted in the first successful mGlu2/3 receptor agonistic medication and is supported by significant findings in glutamatergic system-regulating genes (SLC1A6, SLC1A2, GRIN1, GRIN2A, GRIA1, NRG1, ErbB4, DTNBP1, DAAO, G72/30, GRM3) [2].

The pathophysiology of the condition, which is reflected by the participation of genes including GABRA1, GABRP, GABRA6, and Reelin, has consequently been postulated to be modulated by GABA. In addition, multiple genes, including DISC1, RGS4, PRODH, DGCR6, ZDHHC8, DGCR2, Akt, CREB, IL-1B, IL-1RN, IL-10, and IL-1B, have been identified to be involved in the condition. These genes are implicated in immunological, signaling, and networking deficiencies [2].

There is no specific treatment for schizophrenia. Scientists have the privilege to investigate already developed bio-medical databases using advanced computational tools and techniques, which will lead to the discovery of biomarkers and possible treatments for schizophrenia-related disorders. Mejia et al. have explored the potential to reposition 70 drugs investigated in 231 clinical studies as potential candidates for repurposing pharmaceuticals for schizophrenia based on their interactions with the dopaminergic system. During dynamics simulation analysis, flunarizine-D2-like receptors demonstrated good interaction and stability [3]. A recent computational study found the compound ZINC74289318 as a serotonin 5-HT2A and dopamine D2 inhibitor [4]. Human D-amino acid oxidase (h-DAAO) may effectively act on D-serine, which is being investigated as a possible therapeutic target for the treatment of schizophrenia. 1,2,4-triazine derivatives were discovered to have h-DAAO inhibitory characteristics using 3D-QSAR modeling, docking, and dynamics analysis [5].

Gümüş et al., (2022) conducted combined computational studies and found sulfa drug-pyrrole conjugates as carbonic anhydrase and acetylcholinesterase inhibitors [6]. Another recent quantum computational and spectroscopic study supported by DFT/TD-DFT, molecular docking, and ADMET methods revealed different properties of N-(2-((2-chloro-4,5-dicyanophenyl)amino)ethyl)-4-methylbenzenesulfonamide [7].

The lead compound Martynoside (CID:5319292) has been reported for its antioxidant properties [8], while a comparative toxicogenomic database shows literature-based evidence showing that acteoside (CID:5281800) has a therapeutic role in several diseases such as leukemia [9], inflammation [10], skin neoplasms [11], and wounds and injuries [10]. A previous study suggests that acteoside has antioxidant and neuroprotective activity, and herbs containing it are used to enhance memory [12]. Experimental data reported by Chen et al., (2020) proved that after administering echinacoside and acteoside in a rat model, the typical pathological features of osteoporosis and Alzheimer’s disease were ameliorated [13]. CID:44559250 (Dehydroapateline) is documented to exhibited anti-acetylcholinesterase activity [14].

Unfortunately, a one-size-fits-all natural antipsychotic does not exist. A combination of nutritional recommendations, on the other hand, may help lessen symptoms. However, what helps one person might not always work for the next. Natural compounds’ medicinal properties investigation as a complementary and alternative treatment for schizophrenia could also be executed using computational tools.

The majority of molecular targets for schizophrenia are found in the three major components of the Neurotransmitter release cycle (NRC): serotonin, dopamine, and glutamate, according to studies. Therefore, we have extracted and analyzed all 51 gene interactions from NRC. Natural compounds abundant in *Bacopa monnieri* (*B*. *monnieri*) were chosen to target selected biomolecules to assess the interaction properties of these natural compounds as an alternative treatment for schizophrenia. *B*. *monnieri* is a plant used as a traditional natural medicine in the Asian sub-continent for centuries. *B*. *monnieri* can enhance biochemical molecule activity and protect brain cells, enhancing memory, thinking, and stress reduction could also help treat Alzheimer’s disease [15,16].

Our previous investigation, including molecular interactions between human acetylcholinesterase (AChE) and butyrylcholinesterase (BuChE), enzymes with natural compounds from *B*. *monnieri* revealed that bacoside X, bacoside A, 3-beta-D-glucosylstigmasterol and daucosterol could be suitable cholinesterase inhibitors and need to be tested against neuro disorders through in vivo/in vitro experimentation [17].

Identified potential biomarker STXBP1 biochemical mechanism includes vesicular transportation and neurotransmitter section. A recent study revealed that STXBP1 mutation leads to amyloid-like fibrils in rat brains [18]. Pathogenic STXBP1 variants cause severe early-onset developmental and epileptic encephalopathy [19]. The patient-based study by Stamberger et al., (2016) found that patients with STXBP1 mutation have severe intellectual disabilities [20]. A comparative study between neurodevelopment disorders and intellectual disability patient groups revealed that the STXBP1-associated group had severe global adaptive impairments, fine motor difficulties, and hyperactivity [21].

In this study, we chose natural compounds from *B*. *monnieri* to investigate their inhibitory potency against potential biomarkers of schizophrenia found by pathway analysis and protein-protein interactions (PPI) utilizing a system biology method.

## 2. Material and Methods

### 2.1. Neurotransmitter Release Cycle Biomolecules Data Mining

We obtained information on the 51 NRC genes corresponding to their UniProt IDs from the Reactome database [22] (Appendix A). A computed interaction-based protein-protein network of these 51 gene targets was created after putting UniProt IDs as an input to the STRING (version 11.5) network analyzer [23]. The parameter to the PPI network was set as evidence-based, with the highest confidence level score of 0.9, with 50 interactors in the first and second shells. Generated STRING network was further deeply analyzed by Cytoscape version 3.9.1 [24]. A network analyzer tool was utilized to investigate the topology parameters, e.g., shortest path length, node degree distribution, average clustering coefficient, and average neighborhood connectivity of the PPI network [25].

The network was analyzed using tools integrated into CytoScape, e.g., Molecular COmplex DEtection (MCODE) and GO functional enrichment analysis, ClueGO. Parameters such as the threshold *p*-value were set as <0.05. The output proteins list generated more confident PPIN after setting 50–50 interactors in the first and second shells. The nodes thus obtained from the network were sorted based on their topological properties, e.g., degree, clustering coefficients, betweenness and bottleneck scores [12].

### 2.2. In Silico Interaction Investigation

#### 2.2.1. STXBP1 Preparation as a Receptor Molecule

The 3-Dimensional (3D) crystal structure of Human STXBP1 was not available in Protein Data Bank (PDB). Therefore, it was necessary to develop a 3D model of STXBPI_Human based on the homology modeling approach. SwissModel online workspace [26] was used to generate a 3D model, and further assessment of the model was performed by the MolProbity tool integrated into the SwissModel server [27] (Appendix A).

#### 2.2.2. Active Site Prediction by CASTp Server

The CASTp 3.0 server was used to determine the active site of the STXBP1 modeled structure. The CASTp server, in essence, accepts 3D protein structures in PDB format as input for topographic computing [28] (Appendix A).

#### 2.2.3. Natural Compounds Library Preparation as a Ligand

Structural and chemical information of available natural compounds in *B*. *monnieri* was extracted from PubChem online server, and control drug Quetiapine information was retrieved from DrugBank Database (https://go.drugbank.com/drugs/DB01224 (accessed on 24 August 2022)). A total of 123 natural compound details were extracted, and the *.sdf* library was obtained. The DataWarrior tool was used to predict the compounds’ physicochemical properties, drug-likeness, and drugscore, as well as the mutagenic, tumorigenic, reproductive effect, and irritant properties of 123 natural compounds [29].

#### 2.2.4. Energy Minimization

Receptors and natural compounds need to minimize energy to fix missing and nonstandard amino acid residues and atoms for virtual screening/molecular docking analysis. CHARMm force-field parameters were used for minimization, an integrated tool in Discovery Studio Visualizer 2021.

#### 2.2.5. Molecular Docking Analysis

Molecular interaction analysis was performed by AutoDock 4.2 [30], followed by a combination of empirical free energy force field and Lamarckian Genetic Algorithm (LGA) to build receptor and compound complexes. Using AutoDock utilities, polar hydrogen atoms, Kollman united charges, and salvation parameters were added to the selected receptor molecules. Afterward, the ligand molecules were charged by Gasteiger. Grid box to cover the receptor’s active site was set to X, Y, and Z coordinate of a grid point with needed values accordingly and for grid center with a default value of grid points spacing 0.375, while other parameters were left at their default values.

#### 2.2.6. Molecular Dynamics Simulation (MDS)

Selected complexes based on docking results were also simulated by the GROMACS tool [31] for 100 ns. The receptor molecules’ topology files were built by pdb2gmx tool after implementing the CHARMM27 all-atom force field. In the next step, the control drug and selected natural compounds topology files were created by the SwissParam server [32]. The unit cell triclinic box filled with water was used for the solvation step. The system was stabilized by minimum energy and addition of Na+ and Cl− ions and equilibrated as selected complexes required. It was followed by two-step ensembles NVT (constant number of particles, pressure, and temperature) and NPT (constant number of particles, pressure, and temperature). Both ensembles control temperature and pressure coupling resulting in constancy and system stabilization through complete simulation [33]. GROMACS has many different scripts for complexes MDS analysis and was utilized gmx rms for Root Mean Square Deviation (RMSD) [34], gmx rmsf for Root Mean Square fluctuation (RMSF), gmx gyrate for the calculation of Radius of Gyration (Rg) [35] and gmx H bond for the calculation of numbers of hydrogen-bond (H bond) formed during the interaction.

## 3. Results and Discussion

We conducted a network analysis that included the PPI of the 51 genes (Appendix A) implicated in the NRC in an attempt to identify the putative target gene or protein that would be responsible for the disease’s manifestation in schizophrenia. STRING and CytoScape were utilized to generate and analyze the interaction network.

CytoScape results revealed the top 15 most interacting proteins SNAP25, HSPA8, SYT1, STX1A, VAMP2, SYN1, STXBP1, CPLX1, RAB3A, SLC18A2, SYN2, GAD1, HSP90AA1, SLC17A7, and GAD2 based on different parameters: Betweenness Centrality, Closeness Centrality, Degree, Number of Undirected Edges. All genes showed degree scores ranging from 60 to 84 while Betweenness Centrality ranged from 0.047 to 0.051 (Table 1).

After analyzing the role of identified proteins from the previously published studies, it was found that SNAP25, STX1A, and VAMP2 are components of the SNARE (soluble N-ethylmaleimide-sensitive fusion protein attachment protein receptors) protein family and structurally tail-like small molecules work in an association of other biomolecules. SNAREs and related proteins are essential for vesicle docking, priming, fusion, and neurotransmitter release synchronization [36].

The network was generated by STRING, followed by clustering in three major clusters. Clusters or modules are tightly interconnected nodes in a network that form a dense subnetwork [37]. Selected genes were distributed in the top 3 clusters (Figure 1A). It was observed that cluster third was most dense where Syntaxin-binding protein 1 (STXBP1) was participating (Figure 1B).

STXBP1 interacts with GTP-binding proteins, most prevalent in the brain and spinal cord and substantially enriched in axons, to regulate synaptic vesicle docking and fusion. It is required for neurotransmission and binds syntaxin, a component of the synaptic vesicle fusion mechanism, in a 1:1 ratio. It can interact with syntaxins 1–3 but not syntaxin 4. Mutation of the STXBP1 gene can lead to STXBP1 syndrome, a rare neurodevelopmental disorder caused by heterozygous variants in the STXBP1 gene and characterized by psychomotor delay, early-onset developmental delay, and epileptic encephalopathy [38].

Due to the established data, we chose STXBP1 as a crucial target protein structurally interacting with other genes in the NRC. Previous research suggests that STXBP1, a component of dysbindin, was co-localized with Munc18-1 at presynaptic terminals in primary cultured rat hippocampus neurons. Overexpression of the mammalian homolog of the unc-18 gene (munc18-1) has also been observed in the brains of schizophrenia patients.

The selection of STXBP1 is that all other top-ranked molecules have thread-like structures and perform the activity only when combined. STXBP1 has a more considerable structure than other top-ranked molecules. Additionally, network analysis revealed that STXBP1 forms a denser network interaction than other genes.

We established a 3D structure of STXBP1 to explore the structural features and interaction of selected natural compounds. The FASTA format of the STXBP1 receptor sequences was obtained from the UniProt database (UniProt ID: P61764). SWISS-MODEL server 3D model generation quality assessment data revealed the overall sound quality of the STXBP1 model (Figure 2A) (Appendix A).

In total, 94.55% of peptide residues were in the excellent area of the Ramachandran plot, while only 1.19% were in Ramachandran Outlier and 2.08% in Rotamer Outlier regions, and no bad bonds were found (Figure 2B). The overall MolProbity score was 1.44. The observed GMQE (Global Model Quality Estimate) was 0.86, and QMEANDisCo global was ±0.05, between the standard cut-off value of 0 to 1. Both combined values can estimate the quality of target-template alignment and the template structure (Figure 2C,D) [39].

Further, CASTp server probed the active site residues within a radius of 1.4 angstrom (Å). The resulting computation of the total Richard’s solvent-accessible area is 685.426 Å^2^ and with the volume of 1474.519 Å^3^ (Appendix A).

The physicochemical qualities, drug-likeness, and drug score of the 123 discovered natural compounds from *B. monnieri* and their mutagenic, tumorigenic, reproductive impact, and irritating properties were then evaluated (Appendix A).

The molecular docking of these 123 compounds was conducted against the active site residues of the modeled STXBP1. After docking, ten compounds (Appendix A) with STXBP1 complexes were shown to have better binding energy (−8.3 to −10.3 kcal/mol) as compared to the control drug Quetiapine (−7.18 kcal/mol) (Table 2).

Natural compound CID:5319292 showed the best binding affinity with −10.3 kcal/mol and formed ten h-bonds. After visualizing 2D models of the complex, it was observed that amino acid resides GLN576, ILE259, LEU573, TYR254, SER146, SER149, and THR570 were involved in the formation of hydrophobic interaction. ALA150, LEU138, and TYR140 formed the Alkyl/Pi-Alkyl bond, while ARG39, GLU260, and LYS577 formed the Pi-Cation Pi-Anion bond.

STXBP1_ CID:5281800 complex showed −9.2 kcal/mol binding energy. During the interaction, ten H bonds formation occurred. Amino acid residues involved in hydrophobic interaction were GLU260, ILE259, SER42, LEU280, LYS46, LYS7, GLU283, SER43, LEU138, THR570, HIS571, SER149, THR574, and LEU573 while TYR254, LEU547, and ALA150 formed an Alkyl/Pi-Alkyl interaction (Figure 3).

STXBP1_CID:44559250 complex showed −9.1 kcal/mol binding energy and formed two H bonds during the interaction. Amino acid residues THR574, GLN576, ASP580, LYS584, THR581, SER146, and ILE572, were involved in hydrophobic interaction. Another type of interaction, e.g., residue ALA150, formed Alkyl/Pi-Alkyl bonds while Pi-Cation Pi-Anion bond built by LYS577, GLU260 amino acid residues.

Furthermore, 100 ns long molecular dynamics simulation generated multiple output files containing data of RMSD, RMSF, Rg, and the number of H bond that occurred during MDS. After creating plots from these data sets, it was observed that RMSD values ranged from 0.3 to 0.7 nm (Figure 4A). STXBP1-CID:5281800 complex exhibited the lowest RMSD value, i.e., between 0.3 and 0.4 nm, which was lower than that of the control throughout the simulation; this observation indicates that CID:5281800 has interacted well with STXBP1 with less deviation than other selected compounds. Interestingly, the STXBP1-Quetiapine complex and STXBP1 simulation in water showed almost similar values, approximately 0.45 nm, and a deviation pattern during MDS (Figure 4A).

RMSF plot representing per amino acid residues fluctuation with the observed value ranging from 0.25 to 1 nm (Figure 4B) and remained stable with 0.25 nm except for a few fluctuations. After analysis, the highest fluctuation of the plot was noticed at 500–550 amino acid residues regions, while some other significant fluctuation was observed at 200–230, 260–280, and 300–340 amino acid residues (Figure 4B). Continuous fluctuation at regular intervals shows that amino acid residues present in these regions are also involved in forming H bonds and different interactions during molecular interaction analysis (Figure 4).

The H bond plot representation 1–6 hydrogen bonds formed between receptor-compound interaction during the 100 ns period (Figure 4C). STXBP-CID:5281800 interaction formed 6 H bonds. RoG plot critical assessment is the most important for evaluating the compactness and stability of STXBP1 during the whole simulation period due to the presence of Drugs and selected natural compounds. RoG values for all studied molecules ranged between 2.55 and 2.75 nm. It was also shown that compactness was mainly maintained with a value of 2.25 nm. Compared to STXBP1, STXBP1-CID:5281800 and STXBP1-Quetiapine showed less value with better stability except for instability by STXBP1-CID:5319292 complex (Figure 4D).

Our findings suggest that the identified natural compounds may be effective STXBP inhibitors for the possible treatment of schizophrenia following experimental validation.

## 4. Conclusions

STXBP1 has been identified as a potential druggable target for schizophrenia in this study. The binding pattern of Bacopa’s natural compounds with STXBP1 was also studied, and it was revealed that only a few compounds (e.g., STXBP1-CID:5281800) have significant binding efficacy with the active site of STXBP1. Additionally, *in vivo/in vitro* experimental testing is necessary to confirm the pharmacological efficacy of *Bacopa monnieri* compounds.

## Figures and Tables

**Figure 1 molecules-27-07050-f001:**
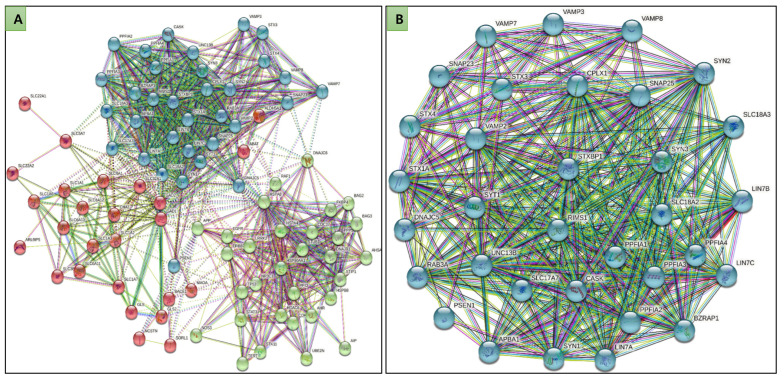
Protein-Protein interaction network of 51 selected genes. (**A**) Top 3 clusters of the interaction, (**B**) showing the largest cluster where STXBP1 interacts with other genes.

**Figure 2 molecules-27-07050-f002:**
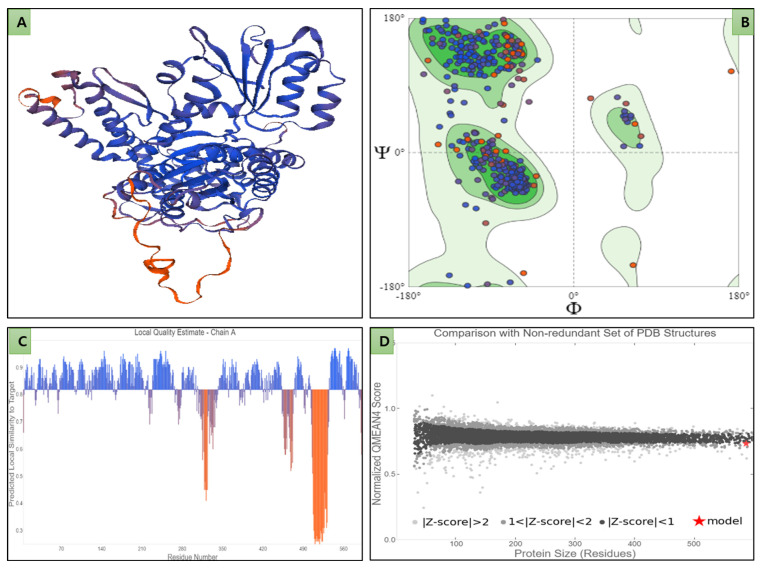
Three-dimensional model and quality assessment of STXBP1. (**A**) Ribbon pattern 3D visualization of 3D STXBP1 model; (**B**) Ramachandran quality assessment plot representing the distribution of peptide residues −phi (Φ) and psi (Ψ) torsion angle; (**C**) local quality estimation plot; (**D**) representing QMean4 score plot of STXBP1 (non-redundant set of PDB structures (★) versus total 3D models available in PDB).

**Figure 3 molecules-27-07050-f003:**
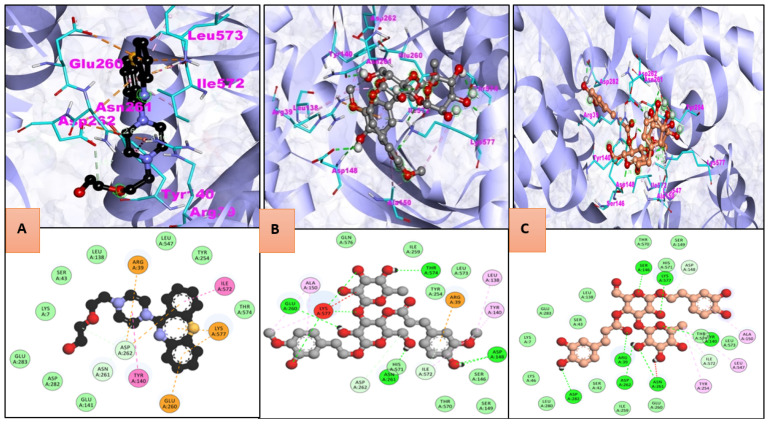
Three-dimensional and 2D interaction of active site residues of STXBP1 with (**A**) control, (**B**) CID:5319292, and (**C**) CID:5281800.

**Figure 4 molecules-27-07050-f004:**
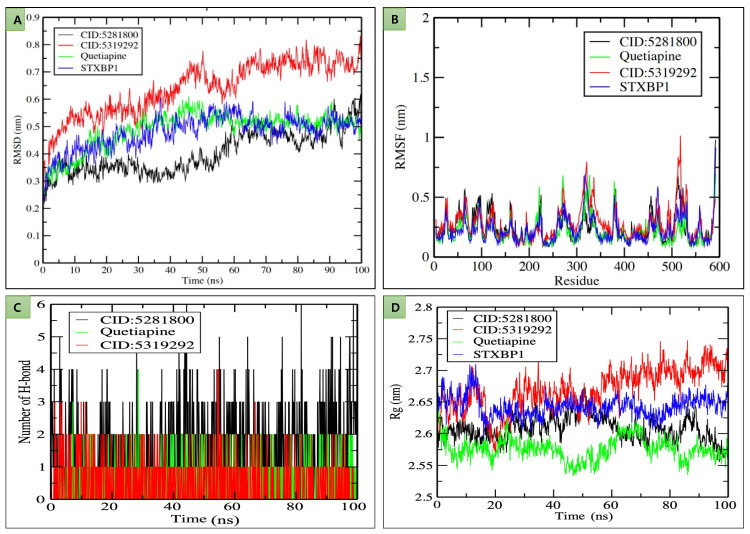
Graphical representation of (**A**) RMSD plot of STXBP1 in water (blue) STXBP1-CID:5281800 (black), STXBP1-CID:5319292 (red), STXBP1-Quetiapine (green) complexes deviation during 100 ns period; (**B**) RMSF plot with fluctuation per residues; (**C**) H bond plot representing the number of H bond during 100 ns period; (**D**) Rg plot representing compactness of receptor molecules in the presence of natural compounds and selected drug during 100 ns simulation. Where nm = nanometer; ns = nanosecond.

**Table 1 molecules-27-07050-t001:** Betweenness Centrality and Closeness Centrality of the 15 selected genes.

S.No.	Name	Betweenness Centrality	Closeness Centrality	Degree	Number of Undirected Edges
1.	SNAP25	0.051693508	0.626761	84	84
2.	HSPA8	0.155216876	0.622378	80	80
3.	SYT1	0.026981174	0.613793	78	78
4.	STX1A	0.027873305	0.605442	76	76
5.	VAMP2	0.027213137	0.605442	76	76
6.	SYN1	0.032573103	0.585526	74	74
7.	STXBP1	0.017975317	0.597315	74	74
8.	CPLX1	0.014901833	0.581699	72	72
9.	RAB3A	0.013181649	0.585526	68	68
10.	SLC18A2	0.035491445	0.597315	68	68
11.	SYN2	0.011852705	0.536145	66	66
12.	GAD1	0.069874322	0.585526	62	62
13.	HSP90AA1	0.032707433	0.523529	62	62
14.	SLC17A7	0.030328683	0.542683	62	62
15.	GAD2	0.047526533	0.570513	60	60

**Table 2 molecules-27-07050-t002:** Molecular docking data of top nine selected natural compounds of Bacopa interaction with STXBP1.

S.No.	Complex Name	Binding Energy (Kcal/mol)	Hydrogen Bonds	H-Bond Length (Angstrom)	Hydrophobic Residues	Alkyl/Pi-Alkyl Residues	Other Type Interaction
1	Drug as a Control Quetiapine	−7.18	ASN261	2.68	GLU141, ASP282, GLU283, LYS7, SER43, LEU138, LEU547, TYR254, THR574	ILE572, TYR140	Pi-Cation Pi-Anion ARG39, GLU260 LYS577
			ASP262	3.56			
			ARG39	3.16			
			LYS577	2.62			
			ASN261	2.68			
			ASP262	3.56			
2	STXBP1_CID:5319292 Martynoside	−10.3	ASP148	2.9	GLN576, ILE259, LEU573, TYR254, SER146, SER149, THR570	ALA150, LEU138, TYR140	Pi-Cation ARG39
			LYS577	2.40			
			LYS577	2.22			
			LYS577	2.94			
			ASP148	2.62			
			THR574	2.23			
			ASN261	2.41			
			GLU260	2.74			
			ASP262	3.56			
			ILE572	2.18			
3	STXBP1_CID:5281800 Acteoside	−9.2	ARG39	1.67	GLU260, ILE259, SER42, LEU280, LYS46, LYS7, GLU283, SER43, LEU138, THR570, HIS571, SER149, THR574, LEU573	TYR254, LEU547, ALA150	NA
			TYR140	3.06			
			SER146	1.77			
			ASN261	2.79			
			LYS577	2.06			
			LYS577	2.14			
			ASP282	2.40			
			ASP262	2.51			
			ILE572	3.71			
			ASP148	2.76			
4	STXBP1_CID:44559250 Dehydroapateline	−9.1	ARG39	2.29	THR574, GLN576, ASP580, LYS584, THR581, SER146, ILE572	ALA150	Pi-Cation Pi-Anion LYS577, GLU260
			ASP151	3.26			
5	STXBP1_CID:5291488 Luteolin 7-galactoside	−8.6	SER146	2.34	THR574, TYR254, LEU138, HIS571, GLN35	ILE572 ALA150	Pi-Cation Pi-Anion LYS577, ARG39 UNFAVORABLE DONOR-DONOR SER149
			SER146	2.03			
			ASP151	2.30			
			ASN261	2.15			
			ASP148	2.58			
			THR570	2.65			
			ILE259	2.31			
6	STXBP1_CID:11145924 Bacopaside C	−8.6	ARG39	2.00	HIS571, SER146, ASP262, THR254, THR574, GLN576	NA	NA
			ALA150	2.34			
			ASN261	2.20			
			LYS577	2.27			
			ASP148	2.49			
			TYR140	2.23			
			ILE259	3.25			
			GLU260	3.55			
			ILE572	3.50			
7	STXBP1_CID:15922618 Bacopaside III	−8.5	ARG39	1.86	LEU547, ILE572, SER146, TYR254, ILE259, GLN576, PRO258, LEU36, GLU260, ASP151, GLN35, ASP148, ALA150, SER149, ASP262, TYR140,	NA	NA
			ASN261	2.143			
			THR574	2.01			
			LYS577	2.16			
			LYS577	2.25			
			ASN261	2.22			
8	STXBP1_CID:11091080 Monnieraside I	−8.3	ARG39	2.70	TYR254, ASP262, TYR140, LEU138, THR570, SER149, ASN261,	Pi-Cation Pi-Anion =GLU260	NA
			SER146	1.86			
			LYS577	2.62			
			LYS577	2.05			
			ILE259	2.09			
			ASP148	2.73			
			ILE572	3.79			
9	STXBP1_CID:9847922 Plantainoside B	−8.3	ASN261	2.60	THR140, ARG39, ASP148, ALA150, ASP580, GLN576, ASP255, THR574	ILE572	Pi-Anion GLU260
			ASP262	1.81			
			LYS577	2.68			
			LYS577	2.95			
			TYR254	2.74			
			ILE259	2.26			
			ILE259	2.75			
			ASN261	2.60			
10	STXBP1_CID:163188454 NA	−8.3	ARG39	2.68	SER149, HIS571, GLN35, TYR140, ILE259, ASP580	ALA150	Pi-Cation Pi-Anion GLU260, ASP148 UNFAVORABLE DONOR-DONOR ASP151, ILE572
			ALA150	2.10			
			ALA150	2.30			
			ASN261	2.67			
			THR574	2.06			
			LYS577	2.24			
			ASP262	3.72			
			TYR254	3.40			
			GLN576	3.68			

## Data Availability

Not applicable.

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
