# Peer review of "Bio-Computational Evaluation of Compounds of Bacopa Monnieri as a Potential Treatment for Schizophrenia"

_molecules, 2022, doi:10.3390/molecules27207050_

Round 1

Reviewer 1 Report

This is an interesting study that explores Bacopa’s natural compounds with STXBP1. This manuscript is interactive and instructive in the search for a natural STXBP1 interactor/inhibitor which could be further utilized in the treatment of neurological disorders.

 Some minor comments need to be addressed to enhance the scientific value of the manuscript prior to acceptance.

Comments/Suggestions for Author:

 1.Introduction section: last paragraph. remove the lines after reference no 8. Because the author has already mentioned Bacopa details with reference numbers 6,8.

 2. Author needs to provide a supplementary file for SWISS-MODEL output.

 3. Which scoring function of the AutoDock 4.2 tool was used? Need to clarify and mention this in the main text.

 4. Section 2.2.5 line 3rd: There are several force fields available in the GROMACS package. Why is CHARMM27 force field used?

 5. Check the spelling of GROMACS in the whole manuscript.

 6. Results and Discussion; Table 1; Did the author check the top-ranking genes/molecules' interaction with bacopa compounds? What was the main analytical reason to choose STXBP1? Need to be clarified.

 7. Table 2: Authors have mentioned the name of a few compounds, it is suggested to identify and include the names of all screened/selected molecules available in the PubChem database.

 8.  Table 2: in the hydrogen bond length; no need to provide length with such precise digits. It needs to only mention as a number after decimal 2 digits. e.g. 2.68023=2.68. correct all accordingly.

 9. The authors should carefully check for typos and punctuation errors throughout the manuscript.

10. Check the reference style as prescribed by the MDPI format.

Author Response

Reviewer 1:

Comments and Suggestions for Authors

This is an interesting study that explores Bacopa’s natural compounds with STXBP1. This manuscript is interactive and instructive in the search for a natural STXBP1 interactor/inhibitor which could be further utilized in the treatment of neurological disorders.

 Some minor comments need to be addressed to enhance the scientific value of the manuscript prior to acceptance.

Comments/Suggestions for Author:

 1.Introduction section: last paragraph. remove the lines after reference no 8. Because the author has already mentioned Bacopa details with reference numbers 6,8.

Reply: Thanks for your concern and suggestion. I have corrected accordingly.

  1. Author needs to provide a supplementary file for SWISS-MODEL output.

Reply: Supplementary file added.

  1. Which scoring function of the AutoDock 4.2 tool was used? Need to clarify and mention this in the main text.

Reply: Thanks for your comment. I have used LGA as a scoring function inbuilt in AutoDock 4.2 tool.

  1. Section 2.2.5 line 3rd: There are several force fields available in the GROMACS package. Why is CHARMM27 force field used?

Reply: CHARMM27 force field

CHARMM27 Force field is most suitable for the solvent model simulations. It include simulation of different chemical,drug like compounds and biomolecules inlusing heterocyclic scaffolds.CHARMM General Force Field (CGenFF)  is a widespread and popular force field for biomolecular simulation, and several recent algorithms such as implicit solvent models have been developed specifically for it.field.https://www.ncbi.nlm.nih.gov/pmc/articles/PMC2888302/

CHARMM have a specific features which can do grid based energy minimization and correction including dihedral angels Ï•, ψ protein backbone dihedrals, as well as all GROMACS features such as virtual hydrogen interaction sites.https://pubs.acs.org/doi/10.1021/ct900549r

  1. Check the spelling of GROMACS in the whole manuscript.

Reply: Corrected accordingly.

  1. Results and Discussion; Table 1; Did the author check the top-ranking genes/molecules' interaction with bacopa compounds? What was the main analytical reason to choose STXBP1? Need to be clarified.

Reply: Thanks for your concern. Yes I have checked the top ranked molecules interaction with bacopa compounds. The reason behind the selection of STXBP1 is that the all other top ranked molecules have thread like structure and they perform activity only when they combine with each other. STXBP1 have bigger structure than other top ranked molecules. Also network analysis revealed that STXBP1 form the more dense network interaction as compared to other genes.

  1. Table 2: Authors have mentioned the name of a few compounds, it is suggested to identify and include the names of all screened/selected molecules available in the PubChem database.

Reply: Thanks for your suggestion. Names have been identified and added accordingly.

  1. Table 2: in the hydrogen bond length; no need to provide length with such precise digits. It needs to only mention as a number after decimal 2 digits. e.g. 2.68023=2.68. correct all accordingly.

Reply: Thanks for your suggestion and corrected accordingly.

  1. The authors should carefully check for typos and punctuation errors throughout the manuscript.

Reply: Thanks for your suggestions. All typographic errors checked and corrected accordingly.

  1. Check the reference style as prescribed by the MDPI format.

Reply: Reference style corrected as per journal requirement.

Reviewer 2 Report

Referee Report

·        Active sites of the proteins should be controlled and some computations should be revised.

·        In Fig. 3, the bond lenghts should be added.

·        Conclusion section should be expanded.

·        Introduction section should be expanded if possible according to related DFT and docking studies such as:

Discovery of sulfadrug-pyrrole conjugates as carbonic anhydrase and acetylcholinesterase inhibitors, Oct 2021, ARCHIV DER PHARMAZIE

Quantum computational, Spectroscopic Investigations on N-(2-((2-chloro-4,5-dicyanophenyl)amino)ethyl)-4-methylbenzenesulfonamide by DFT/TD-DFT with Different Solvents, Molecular Docking and Drug-Likeness Researches, Colloids and Surfaces A Physicochemical and Engineering Aspects 638:128311

·        In docking analysis, why these targets were selected? What is the criteria.

·        The resolutions of the figures should be increased.

·        Conclusion section should be revised.

MINOR REVISION

Author Response

Reviewer 2:

Comments and Suggestions for Authors

Referee Report

  1. Active sites of the proteins should be controlled and some computations should be revised.

Reply: Thanks for your suggestion. I have predicted the active site of the target protein by CASTP tool. More computation details have been added in main text and figure in supplementary file.

  1. In Fig. 3, the bond lenghts should be added.

Reply: Thanks for your valuable suggestion. I have generated new 2D figures with bond length labeling.

3·    Conclusion section should be expanded.

Reply: Thanks for your critical comment. I rewrote the conclusion so that it would be clearer and more precise.

4·    Introduction section should be expanded if possible according to related DFT and docking studies such as:

  1. Discovery of sulfadrug-pyrrole conjugates as carbonic anhydrase and acetylcholinesterase inhibitors, Oct 2021, ARCHIV DER PHARMAZIE
  2. Quantum computational, Spectroscopic Investigations on N-(2-((2-chloro-4,5-dicyanophenyl)amino)ethyl)-4-methylbenzenesulfonamide by DFT/TD-DFT with Different Solvents, Molecular Docking and Drug-Likeness Researches, Colloids and Surfaces A Physicochemical and Engineering Aspects 638:128311 

Reply: thanks for your suggestions. I have expanded introduction with more docking studies examples.

7·        In docking analysis, why these targets were selected? What is the criteria.

Reply: The reason behind the selection of STXBP1 is that the all other top ranked molecules have thread like structure and they perform activity only when they combine with each other. STXBP1 have bigger structure than other top ranked molecules. Also network analysis revealed that STXBP1 form the more dense network interaction as compared to other genes.

8·        The resolutions of the figures should be increased.

Reply: Thanks for your suggestion. I have improved the quality of figures.

10·        Conclusion section should be revised.

Reply. Thanks for your suggestion. I have re-written the conclusion.

Reviewer 3 Report

Dear Author,

I am happy to review your manuscript entitled as: Natural compounds as a potential treatment against Schizo-phrenia: a Bio-computational Approach. The title of manuscript should be revised as your objective is Bacopa plant natural compounds, not overall natural compounds.

I highlithted shortcomings of the article in attached article file but generally manuscript is not well written and must be properly arranged. The major drawback of this study is rational design which is explained below in detail:  

STXBP1 is an important protein that also goes by the name Munc18-1. This protein performs vital roles in the neurotransmission process as discussed by the author in this manucript. There are several diseases linked to this protein and almost all of them are caused by mutations in the gene of this protein, when these mutated proteins are produced in the cell they are either dysfunctional sometimes or sometimes these mutations cause lower expression of this protein which results in different encephalopathies in humans (https://doi.org/10.1080/14728222.2017.1386175) (http://dx.doi.org/10.1212/WNL.0000000000007786). So if there’s any concrete research evidence of overexpression of this STXBP1-protein or accumulation of this protein in the neuronal tissue which causes schizophrenia or other brain-related diseases in humans then this insilico predictive research is well organized and publishable but if there’s no evidence of its overexpression in humans which results in brain-related illnesses the rationale for this research is flawed and it is not acceptable to be published anywhere. If this major problem is addressed then nullify the comments which I provided on the attached manuscript file.

Thanks and regards

Reviewer

Author Response

Reviewer 3:

Comments and Suggestions for Authors

Dear Author,

I am happy to review your manuscript entitled as: Natural compounds as a potential treatment against Schizophrenia: a Bio-computational Approach. The title of manuscript should be revised as your objective is Bacopa plant natural compounds, not overall natural compounds. 

I highlithted shortcomings of the article in attached article file but generally manuscript is not well written and must be properly arranged. The major drawback of this study is rational design which is explained below in detail:  

STXBP1 is an important protein that also goes by the name Munc18-1. This protein performs vital roles in the neurotransmission process as discussed by the author in this manucript. There are several diseases linked to this protein and almost all of them are caused by mutations in the gene of this protein, when these mutated proteins are produced in the cell they are either dysfunctional sometimes or sometimes these mutations cause lower expression of this protein which results in different encephalopathies in humans (https://doi.org/10.1080/14728222.2017.1386175) (http://dx.doi.org/10.1212/WNL.0000000000007786). So if there’s any concrete research evidence of overexpression of this STXBP1-protein or accumulation of this protein in the neuronal tissue which causes schizophrenia or other brain-related diseases in humans then this insilico predictive research is well organized and publishable but if there’s no evidence of its overexpression in humans which results in brain-related illnesses the rationale for this research is flawed and it is not acceptable to be published anywhere. If this major problem is addressed then nullify the comments which I provided on the attached manuscript file. 

Response: Thank you so much for your thoughtful comments and recommendations,, which will undoubtedly improve the quality of this paper.

The role of STXBP1 in neurological disorders is supported by a number of published studies.

For example:

  1. STXBP1 forms amyloid-like aggregates in rat brain and demonstrates amyloid properties in bacterial expression system (Prion. 2021; 15(1): 29–36.)
  2. Munc18-1 is a molecular chaperone for α-synuclein, controlling its self-replicating aggregation (J Cell Biol . 2016 Sep 12;214(6):705-18)
  3. Natural History Study of STXBP1-Developmental and Epileptic Encephalopathy Into Adulthood (Neurology . 2022 Jul 19;99(3):e221-e233.)
  4. STXBP1 encephalopathy: A neurodevelopmental disorder including epilepsy (Neurology. 2016 Mar 8;86(10):954-62).
  5. STXBP1-associated neurodevelopmental disorder: a comparative study of behavioural characteristics (J Neurodev Disord. 2019 Aug 6;11(1):17).

Thanks and regards

Reviewer

All Comments have been extracted from reviewed PDF file,

Comments

1- Revise the title

Reply: Thanks for your suggestion, I have revised the title as “Bio-computational evaluation of compounds of Bacopa monnieri as a potential treatment for Schizophrenia”

2.Systematic

Reply: Thanks for your suggestion. “system biology” is correct at this point.

3.Add complete botanical name.

Reply: Thanks for your suggestion. Added full botanical name.

  1. Write IUPAC names of all compounds mentioned in the manuscript along with structures in the form of table, so structure-activity relationship established, also interactions between compounds functional groups and proteins can easily understand by scientific community to develop future novel anti-schizophrenia drug candidates.

Reply: Thanks for you suggestion. I have added new table with IUPAC names and structures. Please see the supplementary file 4.

  1. Sentence should be revised

Reply: I have revised  as “Further, molecular dynamics simulation methods were used to evaluate selected complexes, and the results show that the STXBP1-CID:5281800 complex has the lowest RMSD value and is highly stable throughout the simulation.”

6.Introduction: Schizophrenia is a ……………risk for the disease.

why this link is here in the manuscript, if it is reference, then include in references  portion.

Reply: Corrected and moved to the reference section.

  1. This is too much long paragraph but without any reference, after three paragraph only one reference is there.

Reply: Thanks for your suggestion. Suitable more references added in the whole manuscript.

  1. Comprehensives literature survey is not carried by respected author

Reply: I have included the appropriate references.

  1. “Earlier studies suggests that most of molecular targets for schizophrenia are from Serotinin, Dopamine and Glutamate Neurotransmitter release cycle (NRC) a three major component of NRC”

Sentence require revision

Reply: It is revised as  “The majority of molecular targets for schizophrenia are found in the three major components of the Neurotransmitter release cycle (NRC): serotonin, dopamine, and glutamate, according to studies.”

  1. References of introduction are not sufficient to support this investigation background

Reply: more relevant references are added

  1. Author should draw structures of these compounds

Reply: These compounds have been described in other publications, which are cited in our manuscript. I value the reviewers' opinion, but it does not appear necessary to draw the structure of these compounds.

  1. No strong rational design to chose Bacopa monnieri plant

Reply: The justification has been described in detail in the introduction and elaborated upon in the discussion.

  1. As a consequence of the established data, we chose STXBP1 as a crucial target protein that structurally interacts with other genes in the NRC

Reference should be cited for this statement

Reply: reference added

  1. Figure 2, Figures quality require improvement

Reply: I have changed the figures with high resolution

  1. “After docking 10 complexes were shown better binding energy (-8.3 to -10.3 kcal/mol) as compared to control drug (-7.18 kcal/mol) (Table 2).”

Name and structure of control must mention in this paragraph

Reply: I have incorporated in the revised manuscript

  1. Table:2

Draw strictures of these nine compounds so that SAR should be established.

Also include SAR portion on the basis of binding energy and interactions with natural compounds functional groups

Reply: as per the suggestions I have drawn the structures. I regret that I am currently unable to include the SAR portion based on binding energy and interactions with natural compound functional groups.

  1. “The lead compound Martynoside (CID:5319292) have reported antioxidant proper-ties [22] While comparative Toxicogenomic database shown literature based evidence shown that

Acteoside (CID:5281800) have therapeutic role in several disease like Leukemia [23] Inflammation [24] Skin Neoplasms [25] wound and injuries [24]. A previous study sug-gests that Acteoside, is known to have antioxidant and neuroprotective activity, and herbs containing it are used to enhance memory [26]. Experimental data reported by Chen et. al., 2020 proved that after administering with echinacoside and acteoside in a rat model, the typical pathological features of osteoporosis and Alzheimer's disease were amelio-rated [27]. CID:44559250 (Dehydroapateline) is documented to exhibited anti-acetylcho-linesterase activity [28].”

This portion will be include in introduction part.

Reply: Thanks for your suggestion. Portion has been moved to introduction.

  1. STXBP1-CID:5281800 complex shown the lowest RMSD value i.e. between 0.3-0.4nm which was significantly also lower than Control STXBP1-Quetiap-ine complex and STXBP1 simulation in water during whole simulation period this observation prove that CID:5281800 have interacted well with STXBP1 with less deviation as compared to other selected compounds.

Improve sentence structure

Reply: it is revised as “STXBP1-CID:5281800 complex exhibited the lowest RMSD value, i.e. between 0.3 and 0.4 nm, which was lower than that of the control throughout the simulation; this observation indicates that CID:5281800 has interacted well with STXBP1 with less deviation than other selected compounds.”

  1. In conclusion, author himself is confused about study outcome, therefore he has to develop rational design of schizophrenia drugs and bioactive derivative with Bacopa monnieri active compounds on the basis of SAR.

Reply: I have rewritten the conclusion section to make more clear as  “STXBP1 has been identified as a potential druggable target for schizophrenia in this study. The binding pattern of Bacopa's natural compounds with STXBP1 was also studied, and it was revealed that only a few compounds (e.g. STXBP1-CID:5281800) have significant binding efficacy with the active site of STXBP1. Additionally, in-vivo/in-vitro experimental testing is necessary to confirm the pharmacological efficacy of Bacopa monnieri compounds”

  1. References

Introduction references are less than required, no proper references are cited to develop rational design to use Bacopa monnieri compounds against schizophrenia disease.

Reply:I have revised as per the suggestions.

  1. References are not cited according to format of journal.

Reply: with the help of Endnote, I have updated the references in revised MS

Round 2

Reviewer 3 Report

Dear Worthy Author,

I am happy that you put lot of effort to improve your manuscript. I recommend the publication of this manuscript after English language, spell and grammatical errors check. 

Thanks and regards

Reviewer

Author Response

Reviewer3

Comments and Suggestions for Authors

Dear Worthy Author,

I am happy that you put lot of effort to improve your manuscript. I recommend the publication of this manuscript after English language, spell, and grammatical errors check. 

Reply: Dear Honorable reviewer, thanks for your suggestions and appreciation, which will definitely improve the overall quality of this manuscript. I have checked the English grammar and error in the whole manuscript. Necessary changes are visible in the track change option incorporated into the Microsoft Word file.
